# Extracellular Matrix Changes in Subcellular Brain Fractions and Cerebrospinal Fluid of Alzheimer’s Disease Patients

**DOI:** 10.3390/ijms24065532

**Published:** 2023-03-14

**Authors:** Lukas Höhn, Wilhelm Hußler, Anni Richter, Karl-Heinz Smalla, Anna-Maria Birkl-Toeglhofer, Christoph Birkl, Stefan Vielhaber, Stefan L. Leber, Eckart D. Gundelfinger, Johannes Haybaeck, Stefanie Schreiber, Constanze I. Seidenbecher

**Affiliations:** 1Leibniz Institute for Neurobiology (LIN), 39118 Magdeburg, Germany; 2Department of Neurology, Otto-von-Guericke University, 39120 Magdeburg, Germany; 3Center for Intervention and Research on Adaptive and Maladaptive Brain Circuits Underlying Mental Health (C-I-R-C), Jena-Magdeburg-Halle, 07743 Jena, Germany; 4Center for Behavioral Brain Sciences (CBBS), 39104 Magdeburg, Germany; 5Institute for Pharmacology and Toxicology, Otto-von-Guericke-University, 39120 Magdeburg, Germany; 6Institute of Pathology, Neuropathology and Molecular Pathology, Medical University of Innsbruck, 6020 Innsbruck, Austria; 7Diagnostic and Research Center for Molecular Biomedicine, Institute of Pathology, Medical University of Graz, 8036 Graz, Austria; 8Department of Neuroradiology, Medical University of Innsbruck, 6020 Innsbruck, Austria; 9Division of Neuroradiology, Vascular and Interventional Radiology, Medical University of Graz, 8036 Graz, Austria; 10German Center for Neurodegenerative Disorders (DZNE), 39120 Magdeburg, Germany

**Keywords:** aggrecan, brevican, cerebrospinal fluid/CSF, HAPLN1, neurocan, tenascin-R

## Abstract

The brain’s extracellular matrix (ECM) is assumed to undergo rearrangements in Alzheimer’s disease (AD). Here, we investigated changes of key components of the hyaluronan-based ECM in independent samples of post-mortem brains (N = 19), cerebrospinal fluids (CSF; N = 70), and RNAseq data (N = 107; from The Aging, Dementia and TBI Study) of AD patients and non-demented controls. Group comparisons and correlation analyses of major ECM components in soluble and synaptosomal fractions from frontal, temporal cortex, and hippocampus of control, low-grade, and high-grade AD brains revealed a reduction in brevican in temporal cortex soluble and frontal cortex synaptosomal fractions in AD. In contrast, neurocan, aggrecan and the link protein HAPLN1 were up-regulated in soluble cortical fractions. In comparison, RNAseq data showed no correlation between aggrecan and brevican expression levels and Braak or CERAD stages, but for hippocampal expression of HAPLN1, neurocan and the brevican-interaction partner tenascin-R negative correlations with Braak stages were detected. CSF levels of brevican and neurocan in patients positively correlated with age, total tau, p-Tau, neurofilament-L and Aβ1-40. Negative correlations were detected with the Aβ ratio and the IgG index. Altogether, our study reveals spatially segregated molecular rearrangements of the ECM in AD brains at RNA or protein levels, which may contribute to the pathogenic process.

## 1. Introduction

Alzheimer’s disease (AD) is a severe neurodegenerative disorder characterized by the deposition of β-amyloid plaques and neurofibrillary tau tangles in the brain. This disorder is the major cause of dementia and has a complex symptomatology with progressing atrophy of the central nervous system. Next to genetic risk factors leading to early onset AD the main risk factor is age. Multiple classification schemes for staging of the disease have been developed, including neuropathological staging according to Braak and Braak [1], henceforth referred to as Braak stages, the CERAD stages (defined by the “Consortium to Establish a Registry for Alzheimer’s Disease”) based on neuritic plaque load [2], or the A/T/N scheme based on the three binarized biomarker categories (A) β-amyloid load, (T) tau biomarkers including phosphorylated tau (p-Tau), and (N) neurodegeneration markers [3]. β-Amyloid (Aβ) and tau peptides as well as neurofilament light chain (NF-L) as neurodegeneration indicators [4] can be detected in the cerebrospinal fluid (CSF) of patients to allow differential diagnosis of dementia types and pathologies, e.g., typical AD, non-AD tauopathies, hippocampal sclerosis or mixed brain pathologies [3].

Spatiotemporal progression analysis based on magnetic resonance imaging revealed the earliest signs of atrophy in the amygdala and hippocampus, followed by middle temporal gyrus, entorhinal, parahippocampal and other temporal cortex areas, then thalamus and striatum and finally frontal, cingular, parietal and insular cortex as well as pallidum [5]. A multitude of molecular and cellular mechanisms preceding neuronal loss have been uncovered and linked to the development of AD, such as inflammatory processes and microglia activation, reactive astrocytes, disturbance of calcium signaling, and synapse dysfunction and loss [6]. Additionally, the neural extracellular matrix (ECM), a macromolecular meshwork of glycoproteins, proteoglycans, link proteins and hyaluronan, was reported to undergo rearrangements in human AD brains as well as in animal models for AD. An early hypothesis of AD development, indeed pointed to a role for glycosaminoglycans and proteoglycans in the pathogenesis [7].

Major elements of adult human neural ECM are the lecticans, including aggrecan, brevican, and neurocan, but also link proteins including HAPLN1/Crtl1, which interconnect proteoglycans and hyaluronic acid to a three-dimensional carbohydrate-rich and negatively charged mesh, filling the extracellular space [8]. Aging-related molecular changes in ECM structure and composition leading to higher tissue stiffness and reduced neural plasticity are thought to be critically involved in neurodegenerative disorders, while intact ECM structures are thought to be neuroprotective [9]. In particular, perineuronal nets (PNNs), elaborate structures of the ECM, that ensheath especially parvalbumin-positive GABAergic interneurons, have been identified as elements of interest in AD research. These rather insoluble aggrecan-rich assemblies are discussed as potential protectors against tau pathology [10]. Suttkus and colleagues reported that the neural ECM can restrict the spreading and internalization of tau aggregates [11]. Interestingly, the clinically licensed anti-AD drug Memantine was shown to increase *Wisteria floribunda* agglutinin (WFA)-binding glycosaminoglycans in the ECM [12] and to ameliorate matrix metalloprotease-mediated degradation of aggrecan [13], moving the neural ECM even more into the focus of interest as a potential progression marker and/or therapeutic target.

However, previous reports on AD-related molecular changes of individual ECM components are in part, contradictory. Accordingly, our knowledge about modifications in composition and integrity of neural ECM in AD brains is still incomplete. In a ‘hypothesis and theory’ article, Scarlett and colleagues claim that the WFA lectin binding, which had been used in many studies on ECM changes in AD, does not allow firm conclusions about the integrity of PNN structures, as reduced binding in AD may merely result from changed sulfation patterns in the carbohydrate structures expressed on ECM proteoglycans, and not necessarily from a disintegration or complete loss of ECM structures [14]. Furthermore, ECM destabilization and disturbed ECM–neuron interaction may also be a consequence of changed solubility or (peri-)synaptic membrane association of matrix proteoglycans, and these effects may vary between brain areas.

To circumvent these problems, in our study we used antibodies to detect several key components of the hyaluronan-binding neural ECM, namely the lecticans brevican, neurocan, aggrecan and the link protein HAPLN1/Crtl1, in subcellular fractions from three different post-mortem brain areas, i.e., hippocampus, temporal and frontal cortex, to provide a comprehensive and more differentiated picture of subcellular ECM rearrangement during AD in the human brain. As all of these molecules had been shown to be associated with synapses (see https://www.syngoportal.org/; accessed on 5 January 2022) [15], we specifically looked into synaptosomal fractions. We combine our protein biochemical data with RNAseq data from The Aging, Dementia and TBI Study (http://aging.brain-map.org/; accessed on 5 January 2022) and we also provide measurements of neural proteoglycans brevican and neurocan and their major proteolytic fragments in human AD and control CSF samples to test if ECM changes within the brain parenchyma are reflected by changes in CSF levels.

## 2. Results

### 2.1. Regulation of Key Components of Neural ECM in AD Brains (Post-Mortem Analysis)

To compare levels of brevican, neurocan, aggrecan and HAPLN1 in subcellular fractions of different areas of AD and control brains we used a total set of 19 brain samples grouped into three diagnostic groups (control, high-grade AD = AD_high_, and low-grade AD = AD_low_) according to clinical AD diagnosis criteria. Groups did not significantly differ with respect to age, sex or post-mortem intervals (all *p* > 0.065; Table 1). Brain areas investigated were the hippocampus, temporal cortex, and frontal cortex, representing sites of early, intermediate, and late neurodegeneration in AD. Subcellular fractions studied are homogenates (Hom) containing the entirety of all ECM molecules, soluble fractions (S2) containing mostly diffuse interstitial ECM components, and synaptosomes (Syn) enriched in (peri-)synaptic ECM. We first compared all three groups (control vs. AD_low_ vs. AD_high_) using a Kruskal–Wallis test and then performed a focused analysis on the extreme groups (control vs. AD_high_) in the S2 fraction as the most informative subcellular fraction using a Mann–Whitney test.

Brevican occurs in the adult human brain as full-length and cleaved isoforms, which after chondroitinase ABC (ChABC) digestion migrate at ~145, ~80, and 50–60 kDa in SDS-PAGE (Figure 1A) [16], and the majority of immunoreactivity is found in the S2 fraction. We compared the levels of total brevican immunoreactivity, full-length brevican and its major 80 kDa proteolytic fragment in all subcellular fractions. Across all three diagnostic groups we found a significant reduction in full-length brevican in the soluble fraction S2 from temporal cortex (χ2 = 9.89, *p* = 0.007). Mann–Whitney tests revealed a significant reduction in the AD_high_ compared to the control group for full-length (U = 3.00, Z = −2.89, *p* = 0.002; Figure 1B) and total brevican in temporal cortex (U = 10.00, Z = −2.08, *p* = 0.040). Normalized full-length brevican levels significantly negatively correlated with Braak stages both in the S2 fraction from temporal cortex (Figure 1C) as well as in the synaptosomal fraction from frontal cortex (Figure 1D). No significant differences between AD diagnostic groups were observed in any of the hippocampal fractions.

Neurocan is a major component of the juvenile brain ECM and occurs in the aging brain mostly as 150 and 130 kDa cleavage products, while the full-length proteoglycan is almost undetectable in aged CNS samples (Figure 2A). No significant effect was found across all three groups, but extreme group comparison revealed a significant upregulation of the major 130 kDa isoform in the S2 fraction from AD_high_ as compared to control samples from temporal cortex (U = 9.00, Z = −2.20, *p* = 0.029; Figure 2B). This cleavage product as well as total neurocan immunoreactivity in the temporal S2 fractions revealed significantly positive correlations with Braak stages (Figure 2C and Figure 2D, respectively). We could not find significant differences between AD and control samples either in the hippocampus or in the frontal cortex (all *p* > 0.072).

The human link protein HAPLN1 migrates as an ~40 kDa immunoreactive band. The Kruskal–Wallis test revealed a significant group effect for HAPLN1 in S2 fraction from frontal cortex (χ2 = 7.87, *p* = 0.020) and the Mann–Whitney test showed a significant upregulation in the AD_high_ compared to the control group in this fraction (U = 8.00, Z = −2.32, *p* = 0.021; Figure 3A), but no significant difference in any of the fractions from temporal cortex or hippocampus. Normalized HAPLN1 levels displayed a significantly positive correlation with Braak stages across all samples (Figure 3C), but only in frontal S2 samples.

Aggrecan, the largest member of the lectican family and a major carrier of the WFA-targeted carbohydrate epitope, is a key component of PNNs. After ChABC treatment it migrates as a stack of bands at approximately 400 kDa in SDS-PAGE. It was shown to be diminished in the middle frontal gyrus (BA9 and 46) of AD patients, correlating with plaque load [17]. 

In the current study, comparison across all three groups showed a significant effect in frontal cortex S2 fractions (χ2 = 6.83, *p* = 0.033) with a down-regulation of total aggrecan immunoreactivity in AD_low_. Furthermore, a significantly positive correlation with Braak stages was detected in the temporal cortex of tissue donors (Figure 4B).

As mentioned above, unexpectedly we did not find any differences in the hippocampus, although this brain area is heavily affected by AD pathology. Although our diagnostic groups did not significantly differ according to the age of donors, they represent a spectrum from 51 to 85 years (see Table 1). Thus, we tested whether hippocampal ECM levels correlate rather with the age of the subjects. We calculated Spearman’s partial correlations corrected for the variable group and observed, indeed, significant negative correlations for full-length brevican in hippocampal homogenate (ρ = −0.528; *p* = 0.024) and soluble (ρ = −0.621; *p* = 0.006) fractions, as well as for the 130 kDa neurocan fragment in homogenates (ρ = −0.534; *p* = 0.022) and S2 soluble fractions (ρ = −0.501; *p* = 0.034; Appendix A). These findings indicate that the concentrations of these proteoglycans in the hippocampus decrease from middle to old age.

### 2.2. Gene Expression Analysis of ECM Components in AD Brains (RNAseq Analysis)

As no major disease-related differences were observed in the homogenate fractions containing all ECM components and their isoforms, we concluded that the regulation observed in soluble or synaptosomal fractions may rather be due to posttranslational regulation of stability or solubility of the ECM molecules. To confirm this assumption, we also investigated available transcript levels for brevican, neurocan, aggrecan and HAPLN1 in AD brains. To this end, we downloaded a comprehensive set of RNAseq data from 107 cases (50 demented and 57 non-demented subjects) from The Aging, Dementia and TBI Study (http://aging.brain-map.org/, accessed on 11 October 2022; Appendix A). In group comparisons between non-demented controls and dementia samples we found significant differences only for HAPLN1 transcript levels in the HC (Figure 5A). Correlation analysis revealed no significant link between dementia diagnosis, Braak or CERAD stages and expression levels of aggrecan or brevican neither in the cortical grey (parietal and temporal) and white matter (frontal) nor in the hippocampus. For hippocampal expression levels of neurocan and HAPLN1 we detected significant negative correlations with Braak stages (Figure 5E,F), which were not reflected by protein level differences in our post-mortem hippocampal samples. Interestingly, in the parietal cortex the correlation of HAPLN1 expression with Braak stages was slightly positive (*p* = 0.045; Figure 5G).

In the search for more ECM-encoding RNAs in the database we also detected the brevican interaction partner tenascin-R (TNR), which showed group-wise down-regulation in hippocampus, temporal cortex and frontal white matter of dementia samples (Figure 5B–D). Furthermore, TNR displayed a significant negative correlation of hippocampal RNA expression with Braak stages (Figure 5H).

Spearman correlation analysis across all RNAseq datasets (Figure 6) showed that total brain RNA levels of neurocan, brevican and tenascin-R significantly positively correlated with brain phospho-tau protein levels, but negatively with the Aβ1-42 level and the ratio of Aβ1-42 to Aβ1-40 (Aβ ratio). In contrast, aggrecan RNA levels significantly negatively correlated with phospho-tau and the tau ratio and had a significant positive correlation with the Aβ1-42 level and the Aβ ratio (Figure 6). Among the ECM components, the cerebral expression levels significantly correlated strongest between the known interaction partners brevican and tenascin-R (Figure 6).

### 2.3. Brevican and Neurocan Levels in the CSF of AD Subjects

As we found differences in the levels of total and cleaved neurocan and of brevican in AD cortex samples, we tested whether those were mirrored by concentration differences in the CSF in an independent cohort of AD patients. As the groups (control vs. AD) in our CSF cohort were not matched for sex and age, with the AD group including a higher proportion of females and higher average age as compared to the control group (Table 2), we performed our analyses on sex- and age-corrected levels of brevican and neurocan. Total concentrations of the two proteoglycans (Table 2) were measured using commercial ELISA kits and a multivariate ANOVA revealed no significant group differences (all *p* > 0.103), thus confirming previous reports in the literature [18,19]. In addition, semiquantitative investigation of brevican and neurocan cleavage products on immunoblots revealed no significant difference between groups, altogether indicating that these measures seem unsuitable as bona fide biomarkers or indicators for AD.

However, to search for interrelationships with other factors, we performed a correlation analysis of CSF proteoglycan concentrations with further CSF parameters and with demographic factors across all subjects, independent of diagnosis (Figure 7). This analysis revealed strong and significant positive correlations between all proteoglycan measures, comprising both total and cleaved gene products, indicating a common transport mechanism from brain parenchyma to the CSF. Furthermore, we noticed significant positive correlations with age, total tau and phosphorylated tau as well as with Aβ1-40 and with the other established neurodegeneration marker NF-L, but not with the AD indicator Aβ1-42 or with the albumin quotient Q_Albumin_, a marker for blood–brain barrier integrity. The Aβ ratio as well as the IgG index, a neuroinflammation marker, however, showed significantly negative correlations with neurocan and brevican amounts in the CSF. On the other hand, progranulin as indicator for microglial activation [20] revealed positive correlations with brevican and its cleaved fragment in our sample. Finally, we found a significant negative correlation of CSF brevican levels with the Mini Mental State Examination (MMSE) test results, again pointing to a link between higher CSF brevican and cognitive decline.

## 3. Discussion

The neural ECM composition and integrity determine important biophysical and cell biological functions as well as neuronal circuit activity in the brain. Here, we show a specific pattern of differences in key ECM components in post-mortem brain samples and CSF of AD patients relative to non-demented controls. Figure 8 summarizes our main results in a graphical scheme. While total protein levels of the ECM molecules in cortex and hippocampus homogenates did not differ between groups in our post-mortem cohort, we found significant AD-related differences in transcript levels of these molecules in the hippocampus and parietal cortex of the RNAseq cohort, pointing to differential regulation of the ECM composition at the translational and posttranslational level. Obviously, this involves differences in the solubility of the ECM elements investigated: HAPLN1, neurocan and aggrecan show higher presence in soluble fractions potentially indicating partial disintegration or loosening of PNNs in the cortex of AD patients. This interpretation is in line with the immunohistochemically documented reduction in aggrecan-positive PNNs in AD cortex [17].

### 3.1. Brevican and Neurocan in AD Brains

Evaluation of RNA data from the Aging, Dementia and TBI Study Consortium revealed that transcript levels of brevican correlate with AD parameters such as p-Tau or the Aβ ratio, whereas at the protein level total brevican immunoreactivity was unchanged in brain homogenates. This latter finding is in agreement with a previous study [21], showing normal brevican distribution in human AD post-mortem brain (Brodman areas 17, 18). In contrast to the study by Lendvai et al. [22], we did also not observe differences in total amounts of any of the cleaved brevican fragments in the brain. However, in the Lendvai study samples were neither ChABC-digested nor adjusted for sex, age, or post-mortem delay, making comparisons difficult. Our finding of a negative correlation between full-length brevican in frontal cortex synaptosomes and Braak stages points to the well-known reorganization of frontal cortex synapses and loss of synaptic integrity across aging and AD stages [23]. This may—at least in part—be mediated by perisynaptic loss of brevican, as this proteoglycan was shown to be a dynamic modulator of excitatory synaptic function, because it is involved in the proper localization and dynamics of AMPA-type glutamate receptors at synapses [24]. Interestingly, the negative correlation with Braak stages is not in agreement with an earlier report about increased synaptosomal brevican levels in the APP/PS1 mouse model of AD [25]. However, in the latter study hippocampal synaptosomes were investigated only in a group-wise comparison in middle-aged mice (up to 12 months) and not according to individual human brain pathology. As the mouse model used was demonstrated to develop no or only late tauopathy after 18 months [26] the comparability to the human post-mortem material is limited.

On the other hand, significantly reduced brevican levels in soluble fractions of temporal cortex samples may rather reflect reduced solubility of the proteoglycan, which could result from its pathological accumulation in insoluble ECM structures such as axonal coats [22]. Conversely, brevican’s close relative neurocan and its major cleavage product were found enriched in temporal cortex soluble fractions. This may result from increased synthesis in astrocytes, potentially induced by Aβ peptides via Sox9 activation, as suggested by Yan et al. [27]. Alternatively, as we did not find any changes in neurocan levels in total homogenates, it may also reflect increased proteolytic cleavage of full-length neurocan and/or higher stability of the 130 kDa N-terminal cleavage product as measured in this study. This fragment comprises the hyaluronan-binding domains and may be less integrated or maintained in ECM complexes under AD conditions, in line with a previous report [17].

### 3.2. HAPLN1 and Aggrecan in AD Brains

The link protein HAPLN1 as the PNN master-organizer and the proteoglycan aggrecan as the largest lectican family member in brain form complexes with hyaluronan [28] and are essential components of PNNs. However, they occur also in the soluble interstitial ECM. In accordance with similar findings in severe AD patients’ brains [22], we found increased HAPLN1 levels in the frontal cortex soluble fraction and positive correlations of aggrecan and HAPLN1 immunoreactivity with Braak stages. Since cortical levels of HAPLN1 transcripts were also found to be elevated, we conclude that expression of this ECM component is upregulated in the cortex under AD conditions. Enhanced HAPLN1 levels in soluble fractions may be an indicator for attenuated formation or impaired maintenance of PNN [29]. In contrast, for aggrecan we found a negative correlation of RNA expression with brain pathology markers in the RNAseq sample. We thus hypothesize that our finding at the protein level may rather result from solubility changes of this very large proteoglycan, similar to the case of neurocan. Additionally, for the well-known lectican interaction partner tenascin-R, a glycoprotein with cross-linking abilities for lecticans [30], the RNA expression levels at least in the hippocampus slightly decrease with disease severity.

### 3.3. Brevican and Neurocan in the CSF: Markers for Dementia, Neurodegeneration, or Just Age?

As the biosynthesis of brevican and neurocan occurs largely in neural tissue and fragments of these proteins have been detected in CSF (http://probe.uib.no/csf-pr/; accessed on 1 September 2022) [31] it is of interest to analyze their concentration in the CSF of patients as a potential proxy of brain levels of these signature ECM components. In agreement with our recent study [32], we show that CSF concentrations of these proteoglycans and their major cleavage products positively correlate with the age of the patients and with Aβ1-40, an indicator for the glymphatic flux from the parenchyma to the CSF [33]. Although we did not see a group-wise difference between control and AD patients, which is in line with earlier studies from other labs [18,19,34], we found clear correlations with tau measures, NF-L, and the Aβ ratio, and—at least for brevican—also with the MMSE score. Altogether, this indicates a link between higher lectican concentrations in the CSF in old age with increased neurodegeneration and with cognitive decline. Along this line, Bader and colleagues showed in a recent proteomic study in three different AD cohorts widespread changes in the CSF protein composition in AD, with the largest cluster comprised of extracellular and ECM proteins [35]. Nevertheless, our data indicate that CSF concentrations of lecticans may rather reflect neurodegeneration in general but are not suitable as AD-specific CSF biomarkers. Unexpectedly, the age effects on brevican and neurocan levels in brain and CSF seem to be opposing, which brings the question of the molecular transport mechanism from the CNS into the CSF even further into focus.

### 3.4. Limitations of the Study

Although within the same size range as in similar studies [10,22], the postmortem brain samples used in this study have a relatively small sample size, which may be not fully representative of the target population of AD cases. Furthermore, postmortem material may be prone to artifacts, but the post-mortem intervals did not differ between groups, and we could also confirm some earlier findings from the literature, e.g., [22], arguing for comparable suitability of the material. Finally, since brain and CSF samples used in this study do not originate from the same individuals but from different cohorts it remains unclear to what extent individual proteoglycan levels in the neural parenchyma and in the CSF correspond with each other. To approach this question, experiments in animal models of neurodegenerative disorders may be a first step.

Taken together, from our study a complex picture of ECM reorganization in AD brains evolves, revealing spatially segregated molecular rearrangements at transcript or protein levels, which may contribute to the pathogenic process and, if so, may represent potential new targets for intervention.

## 4. Materials and Methods

### 4.1. Post-Mortem Brain Samples from Graz

In this study we included post-mortem brain samples from 19 patients comprising 12 clinically diagnosed AD patients (4 with low grade AD = AD_low_ and 8 with high-grade AD = AD_high_) and 7 patients without AD diagnosis and no clinical signs of dementia as controls, all from the Graz brain collection, Medical University of Graz. AD_low_ and AD_high_ groups were defined based on the clinical diagnosis, and Braak and CERAD classification was done in the post-mortem brains. Sex, mean age, post-mortem intervals and cause of death of all subjects as well as statistical comparisons of demographic factors between groups are given in Table 1.

### 4.2. Neurological Assessment of the Patient Cohort from Magdeburg

For the investigation of proteoglycan levels in the CSF we used material from a cohort of 70 neurological patients recruited from the Department of Neurology at the Otto-von-Guericke University Magdeburg between November 2015–November 2019, comprising 35 AD patients and 35 controls. Demographic data are given in Table 2.

AD diagnosis was based on the NINDCS/ADRDA criteria, and the A/T/N classification determined through CSF Aβ ratio, phosphorylated tau (p-Tau) and total tau (t-Tau) [36]. The Mini Mental Status Examination (MMSE) test was performed according to [37]. Cognitively normal control subjects were recruited from a hospital-based cohort of neurologic patients with non-specific complaints who underwent lumbar puncture within a diagnostic workup to rule out any neurologic condition, especially AD. All controls were classified as A/T/N negative in the CSF.

### 4.3. CSF Sample Collection and Biomarker Measurements

For lumbar puncture conducted for diagnostic workup, patients were seated and 9 mL CSF were taken. Relevant contamination of CSF with blood was excluded visually/macroscopically and through microscopy of non-centrifuged CSF aliquots. After centrifugation, the pellet was separated to exclude cells from the analyzed samples. Within 20 min after puncture, CSF samples were centrifuged at 4 °C, aliquoted and stored at −80 °C until analysis. CSF biomarkers of neuroaxonal damage and neurodegeneration were determined immediately after sample collection. Neurofilament light chain (NF-L) and progranulin were measured with commercially available ELISAs (NF-light^®^ ELISA, IBL International GmbH, Hamburg, Germany; Human Progranulin ELISA kit, Mediagnost, Reutlingen, Germany). T-Tau and p-Tau were determined through ELISA (Innotest hTauAg or Innotest p-Tau, Innogenetics, Ghent, Belgium), following the manufacturer’s instructions, see also [38,39]. CSF Aβ peptides 1-40 and 1-42 were assessed with Innotest β-amyloid (1-40) and Innotest β-amyloid (1-42) ELISAs, respectively. The Aβ ratio was calculated as (Aβ1-42/Aβ1-40). Thresholds used for the A/T/N classification of samples were set according to the manufacturer’s correspondence and in agreement with [40,41]: Aβ1-42 < 485 pg/mL; Aβ ratio < 0.069, p-Tau > 70 pg/mL and t-Tau > 350 pg/mL. CSF and serum albumin as well as immunoglobulin G were determined by rate nephelometry (Nephelometer Immage 800, Beckman Coulter, Fullerton, CA, USA). The CSF/serum albumin ratio (Q_alb_ × 10^−3^) was calculated to assess blood–brain barrier integrity, and the IgG-Index (Q_alb_/Q_IgG_) was calculated to assess intrathecal immunoglobulin synthesis as a predictor for neuroinflammatory processes.

### 4.4. Ethics and Informed Consent

The use of human samples in this study was approved by the Ethics Committees of the Otto-von-Guericke-University Magdeburg (Sign 07/17, addendum 11/21) and of the Medical University of Graz (EK28132 ex 15-16). Patients from Germany donating biological material gave written informed consent in accordance with the Declaration of Helsinki. Tissue samples and body fluids from Austrian Patients were only collected if the local authorities requested autopsy. Data and material were coded to guarantee patient anonymity.

### 4.5. Antibodies

Primary antibodies used were rabbit anti-brevican “neo” (Rb399) (custom-made; rat neo-epitope QEAVESE; 1:1000) [42], polyclonal sheep anti-human/rat brevican (AF4009, R&D Systems, Minneapolis, MN, USA, 1:2000), polyclonal sheep anti-rat/mouse neurocan (AF5800, R&D Systems, Minneapolis, MN, USA, 1:1000), polyclonal goat anti-human HAPLN1 (AF2608, R&D Systems, Minneapolis, MN, USA, 1:200), polyclonal rabbit anti-aggrecan (AB1031, Sigma-Aldrich/Merck KGaA, Darmstadt, Germany, 1:500), and the antibodies from the ELISA kits (see below). Secondary antibodies used were peroxidase-conjugated AffiniPure donkey anti-rabbit IgG (H + L; 1:2000 for aggrecan; 1:20,000 for brevican neo), peroxidase-conjugated AffiniPure donkey anti-sheep IgG (H + L; 1:5000), and peroxidase-conjugated AffiniPure donkey anti-goat IgG (H + L; 1:2000); (all Jackson ImmunoResearch, Cambridgeshire, UK).

### 4.6. Enzyme-Linked Immunosorbent Assay (ELISA) Protocols

Commercial ELISA kits (RayBiotech Norcross, GA, USA) (ELH-BCAN-1 and ELH-NCAN-1) were used for the quantitative measurement of total brevican and neurocan in the CSF samples (dilution 1:300 for anti-brevican ELISAs and 1:10 for anti-neurocan ELISAs). ELISA measurements were essentially conducted according to the manufacturer’s instructions and to [32], with slight modifications: To compensate for sensitivity differences in ELISA batches the concentration of the recombinant calibrator protein for the standard curve was adjusted according to OD values obtained (highest standard concentrations were set to 3 ng/mL in brevican ELISAs and 25 ng/mL in neurocan ELISAs; standard dilution series: 6 × 1:2 dilution steps). HRP-streptavidin was used at concentrations of 1:200 for brevican and 1:500 for neurocan.

### 4.7. Brain Fractionation and Immunoblotting

Post-mortem brain samples were thawed on ice, homogenized and subcellularly fractionated as described [43]. In brief, homogenate fractions (H) were centrifuged at 1000× *g* for 10 min at 4 °C, the pellet was rehomogenized and both supernatants were combined and centrifuged at 12,000× *g* for 20 min at 4 °C. The resulting supernatants (S2) were collected and the pellet 2 was resuspended, loaded onto a 0.85/1.0/1.2 M sucrose step gradient and centrifuged at 85,000× *g* for 2 h at 4 °C. Afterwards the synaptosomal fraction (Syn) was collected at the 1.0/1.2 M interface and stored with the H and S2 fractions at −20 °C until further preparation.

Before SDS-PAGE, samples were incubated with ChABC (Sigma-Aldrich, St. Louis, MO, USA) at 1 U/mg at 37 °C for 30 min to digest chondroitin sulfate side chains and afterwards solubilized in 5× SDS loading buffer (250 mM Tris/HCl, pH 8, 50% glycerol, 10% SDS, 0.25% bromophenol blue, 0.5 M DTT) and boiled at 95 °C for 10 min. Samples were separated on 2,2,2-trichloroethanol (TCE)-containing stain-free 5–20% Tris-glycine SDS-PAGE under reducing conditions. Before immuno-blotting protein gels were activated under UV light for 5 min. Protein transfer onto PVDF membranes (Merck Millipore, Burlington, MA, USA), blocking and immunodetection were performed according to standard protocols, see also [32]. Immunodetection was performed using an ECL Chemocam Imager (INTAS Science Imaging Instruments GmbH, Göttingen, Germany). To improve semiquantitative comparability of optical density data from immunoblots a reference sample loaded on all gels was used to calibrate individual blot data and compensate for gel differences. Quantification of band intensities was performed using NHI ImageJ software version 1.52a (US National Institutes of Health, Bethesda, MD, USA). The data were normalized to the means of the control samples.

### 4.8. Analysis of RNAseq Data

RNAseq data were obtained from The Aging, Dementia and Traumatic Brain Injury Study (http://aging.brain-map.org/, accessed on 11 October 2022) developed by a consortium consisting of the University of Washington, Kaiser Permanente Washington Health Research Institute and the Allen Institute for Brain Science, supported by the Paul G. Allen Family Foundation (PIs: Richard Ellenbogen and C. Dirk Keene, University of Washington; PI: Eric Larson, Kaiser Permanente Washington Health Research Institute; and PI: Ed Lein, Allen Institute for Brain Science). The cohort originally comprised 55 TBI cases and 55 control cases matched for sex (32 males and 23 females in both groups), age (89.0 +/− 6.3 resp. 89.0 +/− 6.2 years) and post-mortem intervals (4.6 +/− 1.5 resp. 4.7 +/− 2.0 h). From this cohort 3 cases were excluded by the TBI study and we used the data sets of the remaining 107 cases with clear dementia (50) or exclusion of dementia (57). For details see Appendix A. RNAseq data (normalized FPKM values) of 377 samples for ECM-encoding gene products *BCAN* (brevican), *NCAN* (neurocan), *ACAN* (aggrecan), *HAPLN1* and *TNR* (tenascin-R) were combined with brain protein levels for neuropathological factors (tau, Aβ) and clinical information. The data was analyzed after sorting them by donor and brain fraction, thus comprising 94 hippocampal, 91 parietal cortex, 99 temporal cortex, and 93 frontal white matter samples of 107 donors.

### 4.9. Statistical Analysis of Data

Due to the post-mortem sample characteristics, particularly the relatively small sample sizes of the control and AD groups, non-parametric testing was used to compare levels of brevican, neurocan, aggrecan and HAPLN1 in subcellular fractions of different areas (hippocampus, temporal and frontal cortex) of AD and control brains. We first compared all three groups (control vs. AD_low_ vs. AD_high_) using a Kruskal–Wallis test and then performed a focused analysis on the extreme groups (control vs. AD_high_) in the S2 fraction using a Mann–Whitney test (with the exact sampling distribution of U) [44]. In addition, we calculated Spearman correlations with Braak stages across all samples.

In the RNA cohort we performed group comparisons (control vs. dementia) for the RNA expression data for ECM molecules (aggrecan, brevican, neurocan, tenascin-R and HAPLN1) in different brain areas (hippocampus, parietal cortex, temporal cortex, frontal white matter) using a Mann–Whitney test. Furthermore, we calculated Spearman correlations between these RNA expression data with each other and with diagnosis, Braak stages, p-Tau, tau ratio, Aβ1-42 level, Aβ ratio and CERAD stages across all samples.

In the CSF cohort, groups were not matched for sex and age. Therefore, we performed a rank analysis of covariance following the protocol of Quade [45]. To this end, the data were rank transformed, and gender and age were partialled out of the dependent variables using multiple regression. Standardized residuals of this analysis were used in a multivariate ANOVA with group (control vs. AD) as a factor. To search for interrelationships of the CSF variables with other factors across groups, we performed an exploratory analysis. We calculated Spearman correlation analysis of CSF proteoglycan concentrations (not using the standardized residuals of the previous group analysis) with further CSF parameters and with demographic factors across all subjects, independent of diagnosis (Figure 7).

The level of significance was defined as *p* ≤ 0.05 and all tests were two-tailed. Statistical analysis was performed using GraphPad Prism version 9 (GraphPad Software, Inc., San Diego. CA, United States), SPSS Version 21 (IBM Corp., Armonk, NY, United States) and JASP Version 0.17 (JASP Team, 2023) [46].

## Figures and Tables

**Figure 1 ijms-24-05532-f001:**
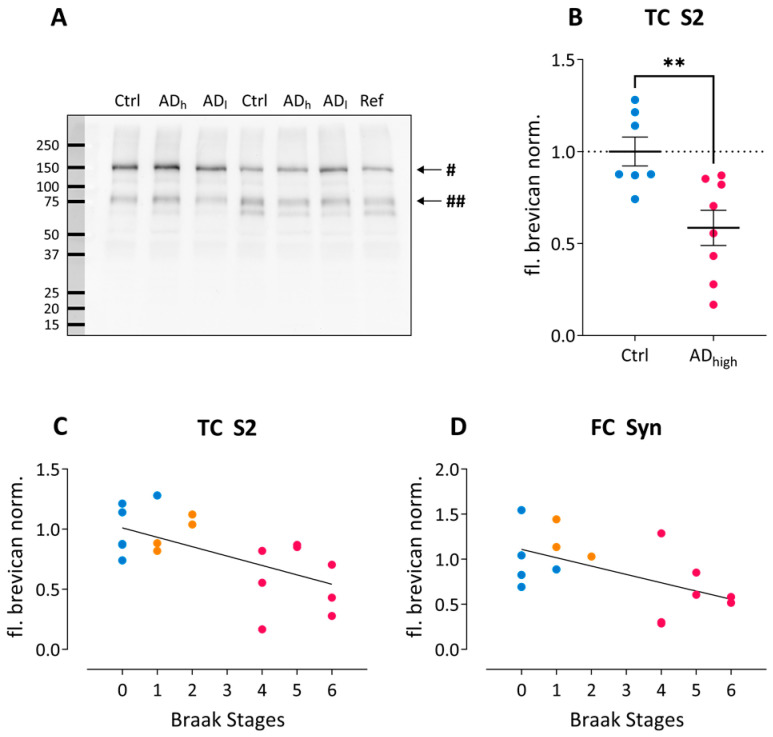
Brevican changes in AD post-mortem brain tissue. (**A**) Example blot showing brevican immunoreactivity in frontal cortex homogenates. Ctrl, Control; AD_l_, low-grade Alzheimer; AD_h_, high-grade Alzheimer; Ref, reference sample. # indicates the 145 kDa full-length brevican; ## marks the 80 kDa C-terminal brevican fragment. (**B**) Group-wise comparison of normalized full-length brevican immunoreactivity in S2 fractions from temporal cortex. ** *p* < *0*.01. (**C**,**D**) Spearman’s correlation analysis reveals significant correlations between Braak stages and normalized full-length brevican immunoreactivity in S2 fractions from temporal cortex (**C**) (Spearman ρ = −0.61; r^2^ = 0.36; *p* = 0.005) and in synaptosomal fractions from frontal cortex (**D**) (Spearman ρ = −0.54; r^2^ = 0.29; *p* = 0.040). Color code in (**B**–**D**): blue, control; orange, low-grade AD; pink, high-grade AD.

**Figure 2 ijms-24-05532-f002:**
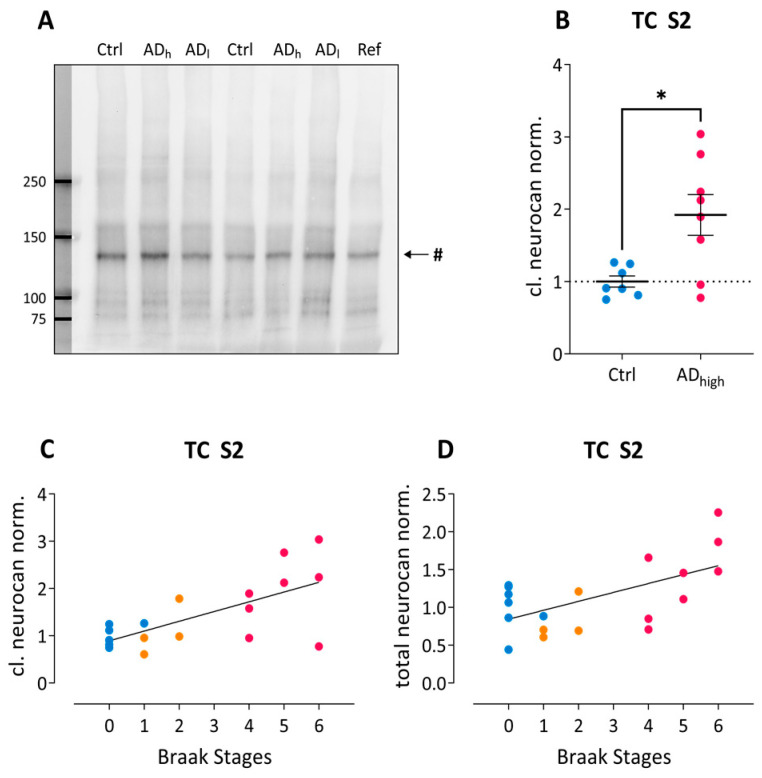
Neurocan changes in AD post-mortem brain tissue. (**A**) Example blot showing neurocan immunoreactivity in temporal cortex homogenates. Ctrl, Control; AD_l_, low-grade Alzheimer; AD_h_, high-grade Alzheimer; Ref, reference sample. # indicates the 130 kDa N-terminal neurocan fragment. (**B**) Group-wise comparison of normalized 130 kDa neurocan immunoreactivity in S2 fractions from temporal cortex. * *p* < *0*.05. (**C**,**D**) Spearman’s correlation analysis reveals significant correlations between Braak stages and normalized 130 kDa neurocan immunoreactivity in S2 fractions from temporal cortex (C; ρ = 0.59; r^2^ = 0.34; *p* = 0.008) and with normalized total neurocan levels in temporal cortex S2 fractions (D; ρ = 0.48; r^2^ = 0.23; *p* = 0.038). Color code in (**B**–**D**): blue, control; orange, low-grade AD; pink, high-grade AD.

**Figure 3 ijms-24-05532-f003:**
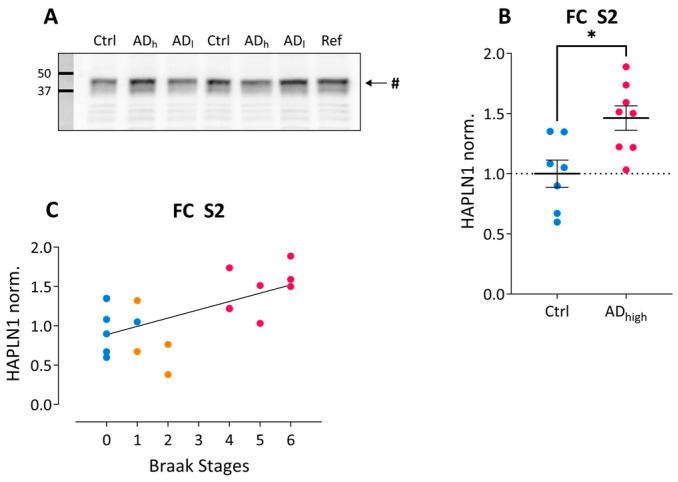
HAPLN1 changes in AD post-mortem brain tissue. (**A**) Example blot showing HAPLN1 immunoreactivity in frontal cortex homogenates. Ctrl, Control; AD_l_, low-grade Alzheimer; AD_h_, high-grade Alzheimer; Ref, reference sample. # indicates the 42–45 kDa HAPLN1 immunoreactivity. (**B**) Group-wise comparison of normalized HAPLN1 immunoreactivity in S2 fractions from frontal cortex. * *p* < *0*.05. (**C**) Spearman’s correlation analysis reveals significant correlations between Braak stages and normalized HAPLN1 immunoreactivity in S2 fractions from frontal cortex (ρ = 0.54; r^2^ = 0.29; *p* = 0.017). Color code in (**B**–**C**): blue, control; orange, low-grade AD; pink, high-grade AD.

**Figure 4 ijms-24-05532-f004:**
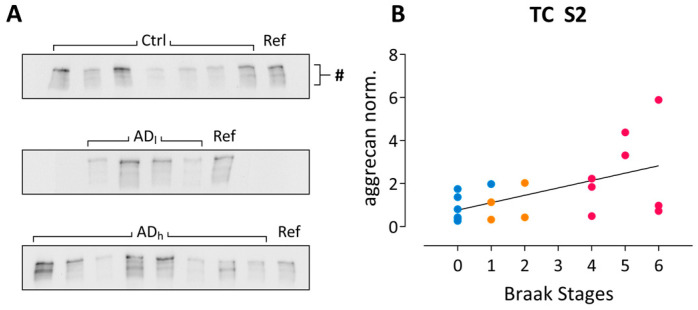
Aggrecan changes in AD post-mortem brain tissue. (**A**) Example blots showing aggrecan immunoreactivity in temporal cortex S2 fractions. Ctrl, Control; AD_l_, low-grade Alzheimer; AD_h_, high-grade Alzheimer; Ref, reference sample. # indicates the approximately 400 kDa full-length aggrecan immunoreactivity. (**B**) Spearman’s correlation analysis reveals significant correlations between Braak. stages and normalized total aggrecan immunoreactivity in S2 fractions from temporal cortex (ρ = 0.49; r^2^ = 0.24; *p* = 0.032). Color code in (**B**): blue, control; orange, low-grade AD; pink, high-grade AD.

**Figure 5 ijms-24-05532-f005:**
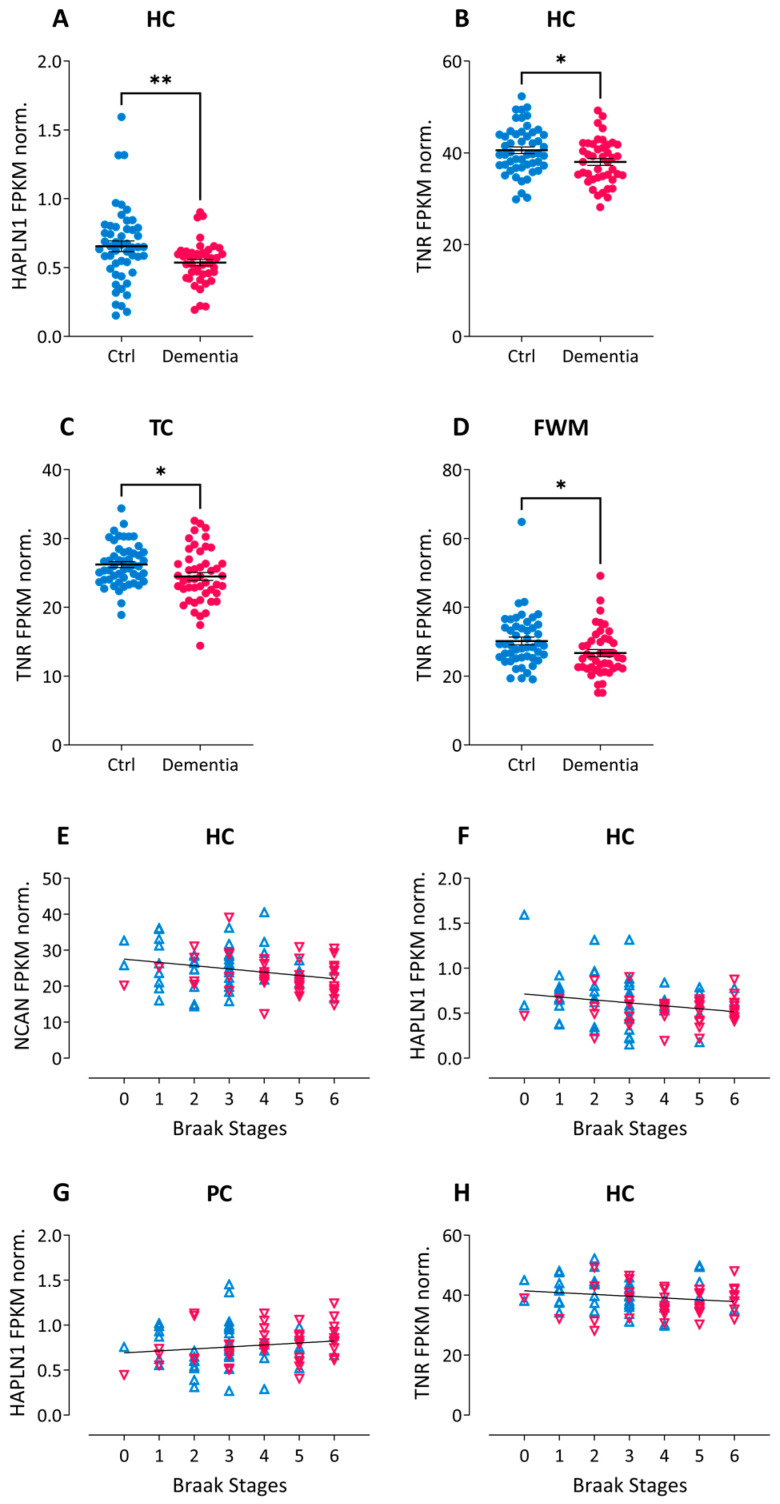
Dementia-related differences in expression levels of ECM components in RNAseq data. (**A**–**D**)**:** Group-wise comparisons in transcript levels of HAPLN1 (**A**) and tenascin-R (TNR; **B**–**D**) between non-demented controls (blue) and dementia samples (pink). * *p* < *0*.05; ** *p* < *0*.01. (**E**–**H**)**:** Spearman’s correlation analysis reveals significant correlations between Braak stages and hippocampal gene expression levels of (**E**) neurocan (NCAN; ρ = -.27; r^2^ = 0.07; *p* =.01), (**F**) HAPLN1 (ρ = −0.20; r^2^ = 0.04; *p* =0.048), and (**H**) tenascin-R (TNR; ρ = −0.22; r^2^ = 0.05; *p* = 0.036) as well as (**G**) parietal expression of HAPLN1 (ρ = 0.21; r^2^ = 0.04; *p* =0.045). Blue triangles: non-demented control samples; pink triangles: dementia samples. All data were obtained from The Aging, Dementia and Traumatic Brain Injury Study (http://aging.brain-map.org/, accessed on 11 October 2022). FPKM norm—Fragments Per Kilobase Million, normalized. HC—hippocampus; PC—parietal cortex; TC—temporal cortex; FWM—frontal white matter.

**Figure 6 ijms-24-05532-f006:**
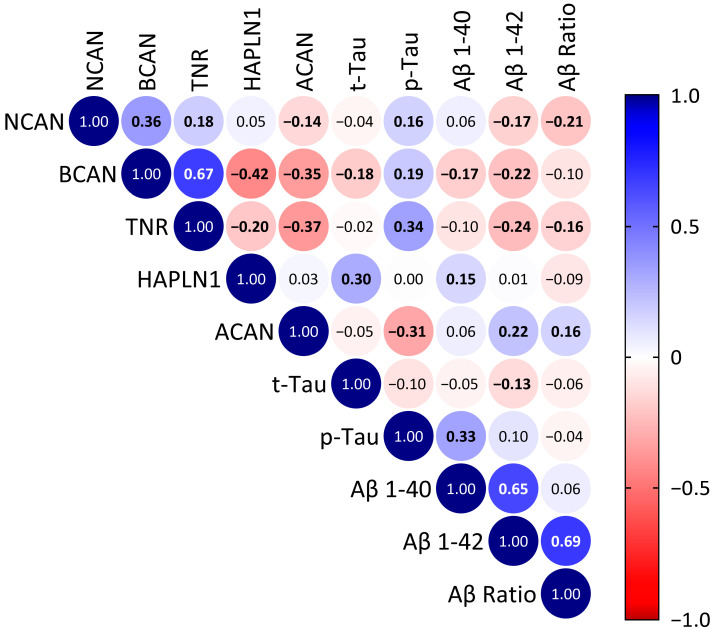
Spearman’s correlation matrix of the RNAseq data for ECM components neurocan (NCAN), brevican (BCAN), tenascin-R (TNR), HAPLN1 and aggrecan (ACAN) with brain pathology markers in the entire dataset. Values show the Spearman rank results (significant correlations are indicated in bold). R values are color-coded with blue colors showing positive correlations, red showing negative correlations. t-Tau, total tau; p-Tau, phosphorylated tau; Aβ 1-40, β-amyloid 1-40; Aβ 1-42, β-amyloid 1-42; Aβ Ratio, β-amyloid(1-42)/(1-40)-ratio.

**Figure 7 ijms-24-05532-f007:**
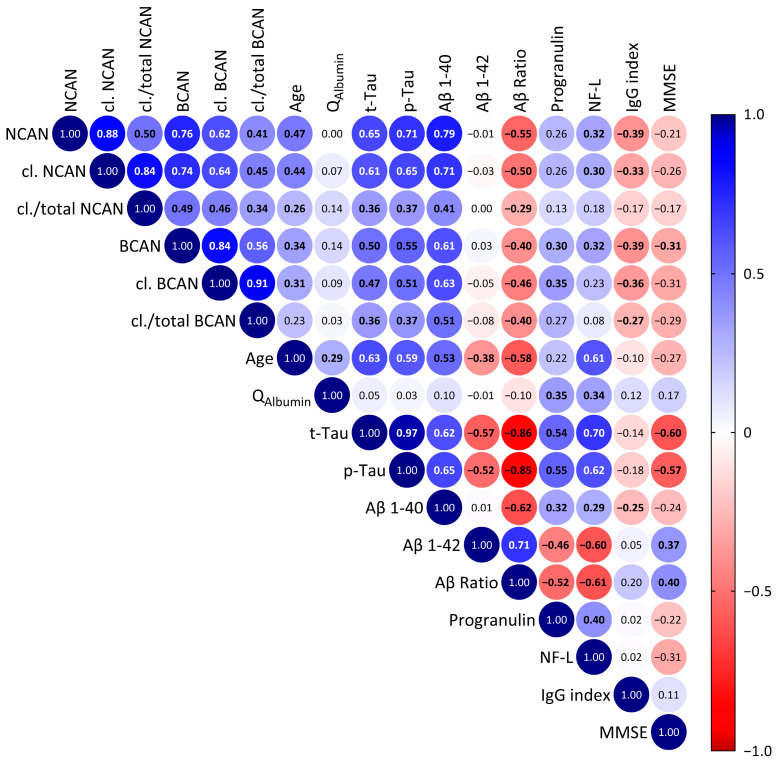
Spearman’s correlation matrix of the concentrations of ECM proteoglycans brevican and neurocan in the CSF of all patients including controls with demographic and serological factors measured during diagnostic work up. Values show the Spearman rank results (significant correlations are indicated in bold). R values are color-coded with blue colors showing positive correlations, red showing negative correlations. BCAN, brevican; NCAN, neurocan; cl., cleaved; Q_Albumin_, albumin quotient; t-Tau, total tau; p-Tau, phosphorylated tau; Aβ 1-40, β-amyloid 1-40; Aβ 1-42, β-amyloid 1-42; Aβ Ratio, β-amyloid(1-42)/(1-40)-ratio; NF-L, neurofilament light chain; IgG index, Immunoglobulin G index; MMSE—mini mental state examination.

**Figure 8 ijms-24-05532-f008:**
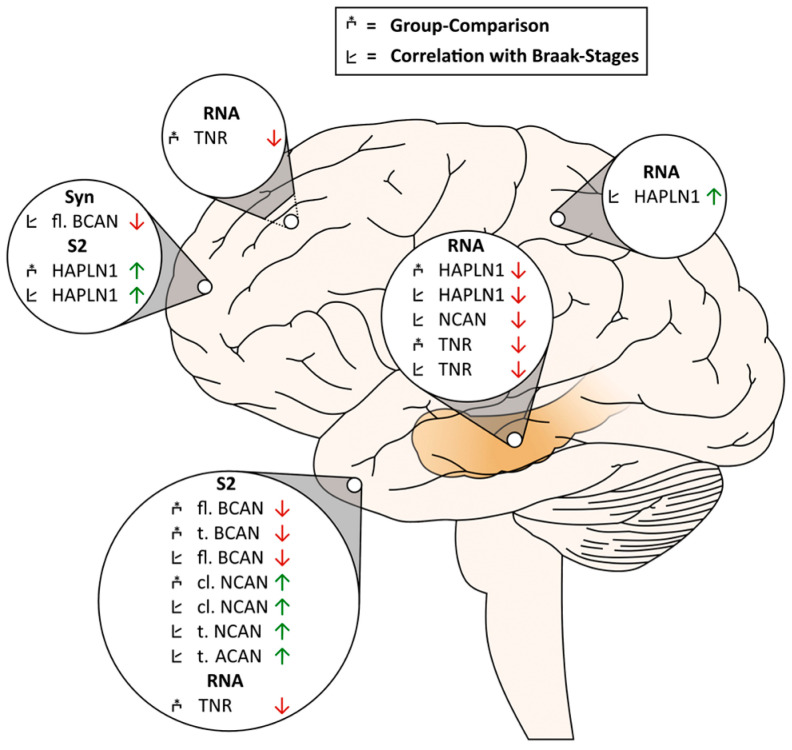
Graphical summary of AD-related changes detected in ECM proteins or RNA levels in the hippocampus, frontal, temporal and parietal cortex. Abbreviations: ACAN—aggrecan; BCAN—brevican; HAPLN1—Hyaluronan And Proteoglycan Link Protein 1; NCAN—neurocan; TNR—tenascin-R; cl—cleaved fragment; fl—full length; t—total. Graphical symbols as specified in the box above the figure indicate if findings are based on group-wise comparisons or on correlations with the Braak stages.

**Table 1 ijms-24-05532-t001:** Demographic data of the post-mortem cohort.

n	Group/Subject	Braak Stage	CERAD Stage	Sex m/f	Mean Age (yrs; ±SD)	Mean PMI (h; ±SD)	Cause of Death by ICD-10 Chapter
7	Controls			4/3	64.0 ± 7.0	15.4 ± 6.1	
	2	0	0	m	59	13.58	Diseases of the respiratory system
	3	0	0	m	60	7.00	Diseases of the circulatory system
	6	1	0	m	61	9.50	Diseases of the circulatory system
	8	0	0	f	78	20.60	Diseases of the circulatory system
	12	0	0	m	60	13.08	Diseases of the respiratory system
	13	0	0	f	61	21.97	Symptoms, signs and abnormal clinical and laboratory findings, not elsewhere classified
	14	0	0	f	69	21.75	Neoplasm
4	AD_low_			2/2	71.3 ± 14.1	12.7 ± 3.6	
	1	2	A	m	83	9.65	Neoplasm
	5	2	A	f	73	17.42	Neoplasm
	7	1	A	m	78	10.05	Symptoms, signs and abnormal clinical and laboratory findings, not elsewhere classified
	11	1	A	f	51	13.50	Disease of the circulatory system
8	AD_high_			5/3	71.4 ± 11.0	20.0 ± 4.5	
	4	6	C	f	80	19.57	Diseases of the circulatory system
	9	4	B	f	70	18.42	Diseases of the digestive system
	10	4	B	m	59	10.27	Diseases of the circulatory system
	15	5	C	m	70	23.42	Symptoms, signs and abnormal clinical and laboratory findings, not elsewhere classified
	16	5	C	m	53	22.83	Diseases of the circulatory system
	17	6	C	m	81	24.33	Diseases of the circulatory system
	18	4	B	m	85	21.75	Diseases of the circulatory system
	19	6	C	f	73	19.50	Diseases of the circulatory system
19	Total cases			11/8	68.6 ± 10.5	16.7 ± 5.6	
χ2				0.17	2.08	5.47	
*p*				0.917	0.354	0.065	

Abbreviations: AD_low_—Low-grade Alzheimer’s Disease; AD_high_—High-grade Alzheimer’s Disease; PMI—post-mortem interval; m—male; f—female; h—digital hours; yrs—years; SD—standard deviation.

**Table 2 ijms-24-05532-t002:** Demographic and biochemical data of the CSF cohort, CSF concentrations of total brevican and neurocan.

	Group	Age (yrs)	Sex (m/f)	Neurocan (ng/mL)	Brevican (ng/mL)	t-Tau (pg/mL)	p-Tau (pg/mL)	Aβ 1-40 (pg/mL)	Aβ 1-42 (pg/mL)	Aβ Ratio	Progr (ng/mL)	NF-L (pg/mL)	IgG Index	MMSE Score
	Controls	63.4± 12.7		32.47 ± 10.28	249.8 ± 94.69	171.8 ± 66.2	38.30 ± 11.51	8054 ± 2371	978.2 ± 243.7	0.126 ± 0.026	0.73 ± 0.15	957 ± 546	0.52 ± 0.04	28.14 ± 0.69
Mean ± SD	AD	76.6 ± 6.4		43.31 ± 12.2	319.5 ± 120.1	799.5 ± 461.2	111.80 ± 45.09	11,284 ± 4185	534.2 ± 149.8	0.050 ± 0.015	0.99 ± 0.23	2017 ± 911	0.51 ± 0.05	19.52 ± 6.44
	Total cases	70.0 ± 12.0		37.89 ± 12.46	284.7 ± 112.9	490.2 ± 456.5	76.15 ± 49.65	9692 ± 3759	753 ± 300	0.088 ± 0.043	0.91 ± 0.24	1575 ± 936	0.51 ± 0.05	21.03 ± 6.72
N of subjects	Controls		17/18	35	35	34	33	34	34	34	16	25	35	7
	AD		7/28	35	35	35	35	35	35	35	35	35	35	33
	Total cases		24/46	70	70	69	68	69	69	69	51	60	70	40

Abbreviations: AD—Alzheimer’s disease; CSF—cerebrospinal fluid; f—female; m—male; Progr.—progranulin; SD—standard deviation; yrs—years.

## Data Availability

The data from the post-mortem and from the CSF cohorts are available on request from the respective institutions. These data are not publicly available due to ethical concerns. RNAseq data are public at http://aging.brain-map.org/.

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
