# Peer review of "Extracellular Matrix Changes in Subcellular Brain Fractions and Cerebrospinal Fluid of Alzheimer’s Disease Patients"

_ijms, 2023, doi:10.3390/ijms24065532_

Round 1

Reviewer 1 Report

The authors compare the level of ECM elements in post-mortem brains, CSFs and RNAseq data in different brain areas in 3 major groups, control; low-level Alzheimer’s diseased, and high-level Alzheimer’s diseased groups. They also correlate the level of ECM elements to the Braak stages.

They found spatial alterations of the ECM elements in the diseased brain samples compared to the control group.

Major comments

1.    Since the severity of Alzheimer’s disease (the Braak stages too) can be correlated with ageing, it would be strongly recommended to repeat the analysis and correlate the results with normal ageing to exclude the changes that happen during normal ageing. The authors mentioned their recent study, but I didn’t find that the age-related changes in ECM elements were discussed adequately. Ignoring the changes that happen during normal ageing could lead to serious misinterpretation of data.

2.    The authors should discuss more the solubility change of the proteoglycans (ref 14), as they mention in the introduction, but it seems that their results also support this.

3. The authors should include more information related to the functionality of the described ECM elements.

Minor comments: 

Abstract: The first sentence says Alzheimer’s dementia, while Intro: Alzheimer’s disease -be consistent

Introduction line 59 – The first couple of sentences are rather poor – AD is more complex and it’s arguable that the major risk factor is age, especially if it is genetic..

Line 66: What do the authors mean differential diagnosis of dementiaS? Different type of dementia? If so, the authors should elaborate more about the dementias, or rephrase that section.

Line 59: familiar vs genetic AD; I’d argue that it always progresses slowly

Line 62: CERAD?

Line 73: please don’t use the word ‘like’- you can use ‘such as’/’for example’ etc. Later in the text, too.

Lines 100-101: reduced binding in AD may

Line 104: “changed solubility or (peri-)synaptic membrane association of matrix proteoglycans” –

Line 160: barely detectable any more? Please rephrase

Line 249 p=0,045 – all previous p values are written : p= .045

Line 324- what are those areas -17 and 18?

Line 405 -why those values were used for thresholds?

Line 515: Therefore -missing ‘e’

Author Response

1. Since the severity of Alzheimer’s disease (the Braak stages too) can be correlated with ageing, it would be strongly recommended to repeat the analysis and correlate the results with normal ageing to exclude the changes that happen during normal ageing. The authors mentioned their recent study, but I didn’t find that the age-related changes in ECM elements were discussed adequately. Ignoring the changes that happen during normal ageing could lead to serious misinterpretation of data.

We are grateful to the reviewer for this comment, because it led us to an unexpected insight (see below).

But first - as the aim of our study was to analyze AD- and neurodegeneration-related differences independent from age, we controlled for age effects in our cohorts, as detailed in the manuscript. In the post-mortem cohort there was no statistically significant age difference between diagnosis groups, neither between all three groups nor between the extreme groups (control and high AD) (see Table 1). Also in the RNAseq cohort, no age differences between dementia and non-dementia groups were detected (see Supplementary Table 1). To make this point clearer, we provide now more information on this cohort in the Materials & Methods section. Finally, in the CSF cohort there was a clear age effect found, as documented in the correlation matrix in Fig. 7. Since these subjects were not matched for age we performed rank analyses of covariance and partialled out the factor age via multiple regression to separate diagnosis and age effects, as explained in the Materials & Methods part. We now add detailed information about age and sex of this cohort in Table 2.

Now to the unexpected insight: based on this comment by reviewer 1, we indeed looked into the data from the post-mortem cohort again. Although there were no significant age differences between groups, the tissue donors represent an age range from 51 to 85 years across groups, and we performed Spearman’s partial correlation analyses controlled for the factor group and found significant negative correlations with age for brevican and neurocan in the hippocampus, which are not related to Braak stages. We show these data now as new Supplementary Figure 1 and mention them in the Results part (p. 8).

2. The authors should discuss more the solubility change of the proteoglycans (ref 14), as they mention in the introduction, but it seems that their results also support this.

We agree with the reviewer that this finding deserves more consideration and we include now a new paragraph in the Discussion section at p. 12 and refer to Crapser et al., 2020 (Ref. 14 is about sulfation pattern differences in proteoglycans, which we did not investigate in our study).

3. The authors should include more information related to the functionality of the described ECM elements.

We provide now more functional background for the individual molecules in the Discussion section (pp. 13-15).

Minor comments: 

Abstract: The first sentence says Alzheimer’s dementia, while Intro: Alzheimer’s disease -be consistent

Done.

Introduction line 59 – The first couple of sentences are rather poor – AD is more complex and it’s arguable that the major risk factor is age, especially if it is genetic..

We edited sentences 2 and 3 accordingly.

Line 66: What do the authors mean differential diagnosis of dementiaS? Different type of dementia? If so, the authors should elaborate more about the dementias, or rephrase that section.

Yes, the ATN classification allows discrimination of different types of dementia-related brain pathologies, like AD, non-AD tauopathies or neurodegenerative pathologies (Jack et al., 2016). We added this information now in the introduction at p. 2.

Line 59: familiar vs genetic AD; I’d argue that it always progresses slowly

We changed the sentence accordingly.

Line 62: CERAD?

CERAD stands for “Consortium to Establish a Registry for Alzheimer’s Disease”. We explain the abbreviation now in the Introduction (p. 2).

Line 73: please don’t use the word ‘like’- you can use ‘such as’/’for example’ etc. Later in the text, too.

Done.

Lines 100-101: reduced binding in AD may

Done.

Line 104: “changed solubility or (peri-)synaptic membrane association of matrix proteoglycans” –

We are not sure about the reviewer’s intention. Shall the quoted sentence be changed? As outlined above, we add now more information to the question of ECM solubility.

Line 160: barely detectable anymore? Please rephrase

We rephrased the sentence to “…while the full-length proteoglycan is almost not detectable in aged CNS samples”

Line 249 p=0,045 – all previous p values are written: p= .045

Done.

Line 324- what are those areas -17 and 18?

We clarify in the text that we refer to Brodman areas 17 - primary visual cortex, and 18 - secondary visual cortex.

Line 405 -why those values were used for thresholds?

Cut-off points were set according to manufacturers’ correspondence and in agreement with the literature. We provide this information now in the Materials & Methods section at p. 16 and cite two references.

Line 515: Therefore -missing ‘e’

Done.

Reviewer 2 Report

The manuscript entitled” Extracellular matrix changes in subcellular brain fractions and  cerebrospinal fluid of Alzheimer’s disease patients” presented well, although I will mainly concentrate on descriptive focus as the study belongs to human subjects. The manuscript can be accepted after addressing the following points.

1)      The section 2.1, Regulation of key components of neural ECM in AD brains (post-mortem analysis), The study includes the small sample size (19 brain samples), which may not be representative of the larger population of AD patients. Additionally, the results of the study are based on post-mortem analysis, which may be subject to various artifacts and limitations. Explain in reply how you can defend such statements.

2)     Figure 2C and D should be clearly presented in the text,also as you mentioned “We could not find significant differences between AD and control samples neither in the hippocampus nor frontal cortex” elaborate this statement also

3)     Line 346-348, Alternatively, as we did not find any changes…. Up to under AD conditions. Explain based on mention references and comment on the reasons also

4)     In minor, you can check the whole manuscript for grammar and spelling.

Author Response

1. The section 2.1, Regulation of key components of neural ECM in AD brains (post-mortem analysis), The study includes the small sample size (19 brain samples), which may not be representative of the larger population of AD patients. Additionally, the results of the study are based on post-mortem analysis, which may be subject to various artifacts and limitations. Explain in reply how you can defend such statements.

We are grateful for this comment and include now a “Limitations of the study” paragraph at the end of the Discussion section (p. 15) to address these critical points.

2. Figure 2C and D should be clearly presented in the text, also as you mentioned “We could not find significant differences between AD and control samples neither in the hippocampus nor frontal cortex” elaborate this statement also

We are not sure what the reviewer is referring to here. We presented Figs. 2C, D and mentioned these findings from the temporal cortex in the results section, p. 6. In hippocampal and in frontal cortex samples neurocan levels did not differ between AD and controls.

3. Line 346-348, Alternatively, as we did not find any changes…. Up to under AD conditions. Explain based on mention references and comment on the reasons also

We add now a sentence of explanation in the Discussion part at p. 14. See also our reply to comments by reviewer 1.

4. In minor, you can check the whole manuscript for grammar and spelling.

Done.

Round 2

Reviewer 1 Report

Thank you for the clarifications and corrections.

I wouldn't start a sentence with an abbreviation as the authors did in Line 57, sentence 2.

Please check the manuscript for punctuation and commas once again.

Author Response

I wouldn't start a sentence with an abbreviation as the authors did in Line 57, sentence 2.

We changed the sentence accordingly.

Please check the manuscript for punctuation and commas once again.

Done.

Reviewer 2 Report

The authors have presented the manuscript well and can be accepted 

Author Response

Thank you.